# Continuous Renal Replacement Therapy in the Critically Ill Patient: From Garage Technology to Artificial Intelligence

**DOI:** 10.3390/jcm11010172

**Published:** 2021-12-29

**Authors:** Sara Samoni, Faeq Husain-Syed, Gianluca Villa, Claudio Ronco

**Affiliations:** 1Department of Nephrology and Dialysis, S. Anna Hospital, ASST Lariana, 22042 Como, Italy; sarasamoni1@gmail.com; 2Department of Internal Medicine II, University Hospital Giessen and Marburg, Justus-Liebig-University Giessen, 35392 Giessen, Germany; Faeq.Husain-Syed@innere.med.uni-giessen.de; 3Department of Health Sciences, Section of Anesthesiology, Intensive Care and Pain Medicine, University of Florence, 50134 Florence, Italy; 4Department of Medicine (DIMED), University of Padova, 35121 Padova, Italy; cronco@goldnet.it; 5Department of Nephrology, Dialysis and Transplantation, International Renal Research Institute of Vicenza (IRRIV), St. Bortolo Hospital, 36100 Vicenza, Italy

**Keywords:** CAVH, CVVH, CVVHD, CRRT, precision CRRT, CRRT machine, CRRT membranes, ultrafiltration, acute dialysis, AKI

## Abstract

The history of continuous renal replacement therapy (CRRT) is marked by technological advances linked to improvements in the knowledge of the mechanisms and kinetics of extracorporeal removal of solutes, and the pathophysiology of acute kidney injury (AKI) and other critical illnesses. In the present article, we review the main steps in the history of CRRT, from the discovery of continuous arteriovenous hemofiltration to its evolution into the current treatments and its early use in the treatment of AKI, to the novel sequential extracorporeal therapy. Beyond the technological advances, we describe the development of new medical specialties and a shared nomenclature to support clinicians and researchers in the broad and still evolving field of CRRT.

## 1. Introduction

Since its origins, continuous renal replacement therapy (CRRT) has appeared as a potential substitute to hemodialysis (HD) or peritoneal dialysis (PD) in critically ill patients with acute kidney injury (AKI), and as a major tool in the treatment of other critical illness [1]. Currently, CRRT is the prevalent acute RRT modality used in Australia and in most European countries, and its use is increasing in the United States [2]. Furthermore, CRRT is used as cardiac, liver and pulmonary support and in septic patients.

The history of CRRT is marked by technological advances linked to improvements in the knowledge of the mechanisms and kinetics of extracorporeal removal of solutes, and the pathophysiology of AKI and other critical illnesses. In the beginning, the components of CRRT machines (i.e., blood, dialysate, replacement and ultrafiltration pumps and the heater) were derived from disposables and devices used for maintenance HD; later, new dedicated equipment for CRRT was designed, thereby improving the safety and performance of CRRT and extending its use in intensive care units (ICUs) [3]. Currently, the last generation of CRRT machines allows the simultaneous support of different organ functions [4].

In this article, we review the main steps in the history of CRRT, from its discovery to its technical evolution, and from its early use in the treatment of AKI to the novel sequential extracorporeal therapy. Beyond the technological advances, we describe the development of new medical specialties and a shared nomenclature to support clinicians and researchers in the broad and still evolving field of CRRT.

## 2. The Discovery of Continuous Arteriovenous Hemofiltration and Its Evolution to the Current Renal Replacement Therapies

In 1971, Lee Henderson described the basis for convective transport in blood purification techniques. Subsequently, in 1974 he described hemodiafiltration combining convection and diffusion. These seminal papers represented the basis for the development of chronic hemodiafiltration by Leber et al. and continuous arteriovenous hemofiltration (CAVH) by Peter Kramer in Goettingen, Germany [1].

CAVH soon became a reliable alternative to HD or PD in critically ill patients, displaying specific advantages and allowing clinicians to manage patients with acute impairment of renal function. Due to the placement of one catheter in the artery and the other one in the femoral vein, and the subsequent arteriovenous pressure gradient, the blood could circulate through the filter and produce ultrafiltrate without the need for pumps [5]. Therefore, the functioning of this system was based on the patient’s blood pressure, ensuring better hemodynamic tolerance in critically ill patients. Moreover, CAVH had the advantage of simplicity and did not require special equipment, thereby making it possible to perform RRT even in ICUs not fully equipped or trained for HD (Table 1). After the initial resistance to the application of extracorporeal circulation in intensive care, this technique soon became the most commonly used option for critically ill patients.

In 1982, the use of CAVH in Vicenza was extended for the first time to a neonate with the application of specific minifilters [6]. Two years later, CAVH began to be used to treat septic patients, burn patients and patients after transplantation and cardiac surgery, even with regional citrate anticoagulation [7]. At first, in CAVH, the prescribed ultrafiltration rate was achieved manually by arranging the filtrate bag at the right height, thereby changing the negative pressure caused by the filtrate column. The replacement fluid was also regulated manually [8].

Meanwhile, clinical and technical limitations of CAVH spurred new research and the discovery of new treatments, leading to the development of continuous veno-venous hemofiltration (CVVH), continuous veno-venous hemodialysis (CVVHD) and continuous veno-venous hemodiafiltration (CVVHDF). The advantages and disadvantages of CAVH are shown in Table 1.

The low depurative efficiency was overcome by applying filters with two ports in the dialysate/filtrate compartment and through the use of counter-current dialysate flow, allowing the addition of diffusion and the birth of continuous arteriovenous hemodiafiltration or hemodialysis (CAVHDF or CAVHD) [9]. These techniques allowed clinicians to treat patients with higher levels of urea by increasing the dialysate flow rate [10]. Nonetheless, the problems of arterial cannulation and low blood flow rate still remained the main limitations of these techniques.

Thanks to the development of double-lumen venous catheters and peristaltic blood pumps, in the mid-1980s, CVVH was proposed [11]. The presence of a pump that generated negative pressure in part of the circuit made it necessary to add a device to detect the presence of air and a sensor to monitor the pressure in the circuit, to avoid, respectively, air embolisms and circuit explosion in case of coagulation or obstruction of the venous line. Later, ultrafiltrate and replacement pumps and a heater were added to the circuit. Both CAVH and CVVH underwent several technological evolutions to improve the performance and safety of extracorporeal treatments in the ICU (Table 1).

The development of CVVH allows to increase the exchange volumes, and subsequently, the depurative efficiency [12]. The use of counter-current dialysate flow led to further improvements and the birth of CVVHD and CVVHDF.

## 3. The Integrated Technology

When RRT emerged, all the components (i.e., blood, dialysate, replacement and ultrafiltration pumps and the heater) came from machines used in nephrology units to perform maintenance HD; they were not integrated, and therefore were unable to work well together, thereby increasing the risk of errors and technical complications. This situation pushed relevant companies to integrate all these components into machines for CRRT: B. Braun generated the compact ECU Carex machine, Baxter the integrated BM 25 and Hospal the DM 32. Medica manufactured an integrated version of the Equapump, and Equaline, Fresenius Medical Care, built the DM08. Bellco adjusted the Multimat B acute version machine, and Gambro a particular version of the AK 10 module for acute RRT [1,3]. These machines provided the basis for designing new dedicated equipment for CRRT.

A crucial step was reached in the early 1990s, with the introduction of the first integrated CRRT platform explicitly designed for acute RRT in intensive care, i.e., PRISMA^®^. It had four pumps, a pre-assembled circuit and an auto-priming feature, which improved the safety and performance of CRRT, especially with non-expert workers, thereby spreading its use to almost every ICU [13]. In the following years, the continuous and effective collaboration with manufacturers and industrial designers led to the development of new dedicated machines with additional features, easy-to-use friendly interfaces, greater accuracy and superior reliability. All these advances made it possible to prescribe and deliver CRRT safely and effectively. The CRRT equipment, together with disposables and solutions, have been optimized. High quality standards and a significant degree of technological sophistication have been achieved (Figure 1).

One small step in the universe of technology, but at the same time enormous progress for the patients, has been taken in the area of pediatric intensive care. In response to the inadequate and unreliable adopted technology used for small babies and neonates in the past, we saw the birth of CARPEDIEM (Cardio Renal Pediatric Dialysis Emergency Machine)—CRRT equipment specifically designed to respond to the needs of neonates and small children [14] (Figure 2).

## 4. Multidisciplinarity and Clinical Evidence Supporting the Evolving Technologies

In the seventies, nephrology and critical care were two different specialties separated by cultural barriers and education curricula. CAVH represented the common ground for a closer interaction and the birth of a new cooperative effort leading to Critical Care Nephrology being a new branch of medicine [15]. This became known as “Vicenza Model” from the original center where this cooperation was implemented and developed. This integrated multidisciplinary approach to Critical Care Nephrology is rapidly becoming a standard of practice. Nonetheless, the nephrologist’s or intensivist’s perspective, even supported by the related scientific societies, still predominates in several countries [16].

In the nineties, with the gradual emergence of the concept of evidence-based medicine, all these new technologies and the related medical practices required supporting evidence. Therefore, the Acute Dialysis Quality Initiative (ADQI) consensus group was created to provide clinical evidence for the growing techniques [17]. The enormous contribution of the ADQI promoted consensus in various areas of critical care nephrology, such as definition and classification of AKI; the use of AKI biomarkers [18]; lung–kidney, liver–kidney and heart–kidney interactions [19,20,21]; quality standards and quality indicators for an adequate CRRT [22]; prescription and delivery controls through new technological tools (i.e., automatic biofeedback) embedded in the modern dialysis machines; extracorporeal fluid management [23]; application of CRRT in a specific group of patients and clinical areas (cardiac surgery, sepsis, viral infections, etc.) [24,25].

The advancements in critical care and the associated technology has allowed clinicians to treat increasingly ill and comorbid patients. These efforts gathered different specialists to the patient’s bedside (intensivists, nephrologists, cardiologists, trained nurses, etc.) and led to many clinical investigations. Therefore, for clinical and experimental reasons, homogeneous terminology was advocated for. In 2015, a multidisciplinary panel (the Nomenclature Standardization Alliance) standardized definitions, components, techniques and operations of the extracorporeal therapies, thereby generating a standardized nomenclature for manufacturers, CRRT experts, new machine software and clinical trials [26,27].

With the progress in the understanding of the pathophysiology of AKI, new guidelines were developed driving indications, modalities of prescription, monitoring techniques and quality assurance programs [28]. In the meantime, newer criteria for adequacy of treatment were implemented in clinical routine, avoiding under-dosing of treatment and favoring the achievement of therapy targets such as fluid balance control, restoration of homeostasis and correction of biochemical derangements with excellent blood purification, thereby making CRRT a well-established form of therapy in intensive care [29].

Information technology and precision medicine have recently furthered the evolution of CRRT, providing the possibility of collecting data in large databases and evaluating policies and practice patterns [30,31]. The application of artificial intelligence and enhanced human intelligence programs to the analysis of big data has further moved the front of research ahead, providing the possibility of creating silica-trials and finding answers to patients’ unmet clinical needs [32]. The opportunity to evaluate the endophenotype of the patient makes it possible to adjust treatments and techniques by implementing the concept of precision CRRT. This allows clinicians to normalize outcomes and results among different populations or individuals and establish optimal and personalized care.

## 5. Extracorporeal Organ Support 

With the development of new membranes, new techniques, new treatment modalities and finally new solutions, including the introduction of citrate anticoagulation, the entire field of critical care nephrology moved forward, expanding the areas of application of extracorporeal therapies to cardiac, liver and pulmonary support.

In the early 2000s, thanks to the third generation of CRRT machines, equipped with particular circuits to support the functions of organs other than the kidney and with specific devices and biomaterials, the use of extracorporeal therapies was extended to other states of critical illness, such as sepsis and multiple organ dysfunction syndrome [33]. As multiple organ support and sepsis therapy generally require the removal of larger molecules than traditional RRT (e.g., cytokines and inflammatory mediators), treatments with higher doses (i.e., high-volume hemofiltration, HVHF) were prescribed [34,35] and higher permeability membranes (i.e., high cut-off CVVHD, HCO-CVVHD) were developed [36,37,38]. The rationale beyond these techniques was the hypothesis that unselective removal of chemical mediators leads to a reconstitution of immune homeostasis in septic patients (i.e., peak concentration hypothesis) [39].

Additionally, sorbents, which remove other molecules using their absorption capabilities, can be used. Natural compounds, such as coil and aluminosilicate minerals, were known since the 1950s for their capacity to remove harmful substances through chemical and physical bonding. Later, synthetic polymers were developed with this aim. These compounds, assembled in special devices or cartridges and applied in extracorporeal circuits, led to the development of hemoperfusion. While it was burdened with severe adverse reactions in early use due to the poor purity of the sorbents, with the development of synthetic compounds, biocompatibility and tolerance significantly improved. Currently, several hemoperfusion devices are used for different purposes, including the removal of endotoxins. Cartridges with polymyxin-B-coated polystyrenic fibers that actively adsorb circulating endotoxin have been used to treat sepsis. They have demonstrated beneficial effects, especially in abdominal septic shock [40,41]. Sorbents in the treatment of sepsis are also utilized in coupled plasma filtration adsorption (CPFA) treatment, where plasma, previously separated from blood cells, circulates through a cartridge, thereby do not exposing blood cells and platelets to a sorbent that is effective but scarcely biocompatible [42]. Other sorbents, assembled in special filters or cartridges, are currently used for the treatment of sepsis (oXiris, CytoSorb, etc.), although robust evidence in this field is still lacking for most long-term outcomes (e.g., 90-day mortality rate) [43,44].

Furthermore, extracorporeal therapies allow to support or partially replace the functions of organs other than the kidney (i.e., the liver, the lung and the heart).

A new approach in multiple organ dysfunction syndrome related to septic or endotoxic shock is represented by “sequential extracorporeal therapy,” where different techniques, membranes and/or cartridges are sequentially used for a single patient to modulate sepsis and replace or support the functions of the impaired organs [45].

For all these purposes, machines have been modified with specific circuits and adequate software. Hence, 50 years after the first experiments of Henderson et al., we are facing today an entirely new approach to the use of extracorporeal therapies, and the fourth generation machines have become platforms to perform a spectrum of therapies under the umbrella term of extracorporeal organ support (ECOS) [4].

## 6. Conclusions and Future Perspectives in Extracorporeal Blood Purification

Several therapeutic options and new devices are available today, and they represent opportunities for new investigations and research. Biomaterials, co-polymers, sensors and spinning and miniaturized technologies are all areas characterized by considerable multidisciplinary translational research. Similarly, innovative research methodologies, registries and biobanks applied to critical care nephrology will guarantee more accurate results from clinical research, specifically for both short and long-term outcomes of critically ill patients with AKI undergoing extracorporeal treatments. Biomaterials and membrane surface modification and functionalization will undoubtedly lead to improved care thanks to better biocompatibility and less thrombogenicity, likely making possible treatment without anticoagulation [46]. New sorbents may represent a further development for coping with the new challenges of the future. Software and hardware integration with the harmonization of nomenclature and operations will allow developing unified platforms capable of performing any extracorporeal therapy from ultrafiltration to extracorporeal membrane oxygenation. Online sensors will further assist in highly tolerated extracorporeal circulation; controlling blood volume and thermal balance variations; and optimizing fluid status and hemodynamics [47]. Chemical sensors for acid–base balance and electrolytes may provide the basis for continuous adjustments of dialysate and replacement fluid composition.

Finally, miniaturized, wearable and possibly implantable devices to monitor and treat the critically ill patient requiring blood purification [48] in intensive care represent the final frontier, “to boldly go where no man has gone before”.

## Figures and Tables

**Figure 1 jcm-11-00172-f001:**
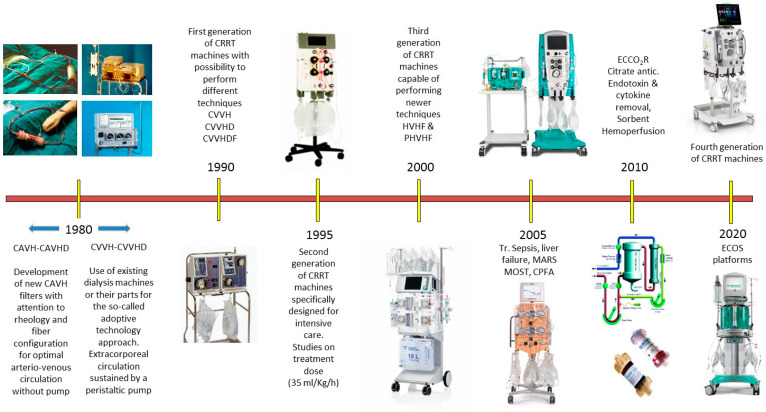
Evolution of continuous renal replacement therapy (CRRT) technology for adult patients (reproduced and modified with permission from Ronco C: Nefrologia Critica, Piccin Nuova Libraria, Padova, 2021). CVVH, continuous veno-venous hemofiltration; CVVHD, continuous veno-venous hemodialysis; CVVHDF, continuous veno-venous hemodiafiltration; HVHF, high-volume hemofiltration; PHVHF, pulse high volume hemofiltration; ECCO_2_R, extracorporeal CO_2_ removal; CAVH, continuous arteriovenous hemofiltration; CAVHD, continuous arteriovenous hemodialysis; MARS, molecular adsorbent recirculating system; MOST, multiple organ support therapy; CPFA, coupled plasma filtration adsorption; ECOS, extracorporeal organ support.

**Figure 2 jcm-11-00172-f002:**
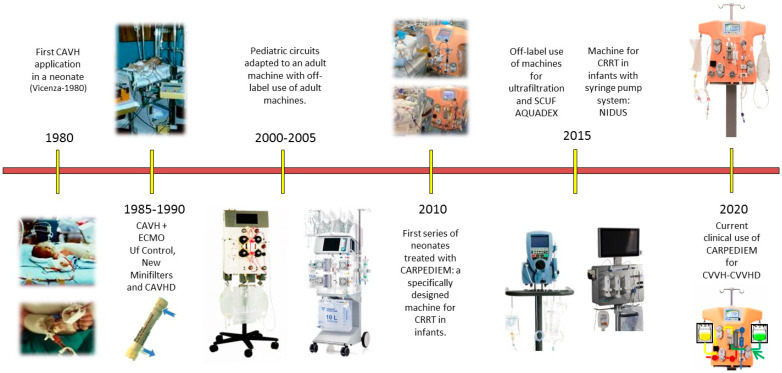
Evolution of continuous renal replacement therapy (CRRT) technology for infants and neonates (reproduced and modified with permission from Ronco C: Nefrologia Critica, Piccin Nuova Libraria, Padova, 2021). CAVH, continuous arteriovenous hemofiltration; SCUF, slow continuous ultrafiltration; ECMO, extracorporeal membrane oxygenation; Uf, ultrafiltration; CAVHD, continuous arteriovenous hemodialysis; CARPEDIEM, cardio renal pediatric dialysis emergency machine; CVVH, continuous veno-venous hemofiltration; CVVHD, continuous veno-venous hemodialysis.

**Table 1 jcm-11-00172-t001:** Advantages and disadvantages of continuous arteriovenous hemofiltration (CAVH) and evolution to continuous veno-venous hemofiltration (CVVH).

Advantages of CAVH	Disadvantages of CAVH	Improvement of CAVH	Improvement of CVVH
Easy and feasible everywhere	Less efficient than HD	Optimization of ultrafiltration by positioning the filtrate bag in a sloping position	Use of progressively more precise blood and ultrafiltrate pumps to increase safety, up to the development of complete machines for CRRT
No blood pump required	Complications related to the arterial cannulation	Optimization of blood flow rate by developing new catheters and shorter lines to reduce resistance	Optimization of blood flow rate (>150 mL/min) by developing double-lumen venous catheters with an adequate caliber
Continuous and physiologic fluid removal	Potential fluid balance errors	Optimization of filtration fraction and gravimetric control of the ultrafiltration	Optimization of hemodynamic tolerance with replacement fluids containing bicarbonate and accurate systems of fluid balancing
Better hemodynamic tolerance than HD	Low depurative efficiency	Introduction of dialysate thus allowing the addition of diffusion to increase the depurative efficiency	Optimization of membrane permeability by using polysulfon, polyamide and polyacrylonitrile, thus increasing cut-off values up to 50,000 Da
		Optimization of filter geometry and development of filters of adequate size for arteriovenous circulation	New anticoagulation strategies and dialysis fluid heating systems

HD, hemodialysis; CRRT, continuous renal replacement therapy.

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
