# Peer review of "Continuous Renal Replacement Therapy in the Critically Ill Patient: From Garage Technology to Artificial Intelligence"

_jcm, 2021, doi:10.3390/jcm11010172_

Round 1

Reviewer 1 Report

Interesting paper resuming the history and developement of continous therapies for AKI in CCU. Would be a reference paper for citation in future Reviews or position paper in this field. I consider as "special article" since it do no brings relevan results but interesting Review of the evolution of the technology tan Young generation tend to forget. Main author seems to be the most indicated person to do it due to his trajectory and experience in this field.I have really enjoyed discovering details that I had forgotten about this technique story.

1.-Due to its structure as a narrative review, I have nothing to add in terms of methods and information. The reference to the need for integrated work between Nephrologists and CCU specialists seems especially relevant to me. This model is unfortunately not common to all countries, which is why other recent position papers of scientific society are especially relevant and could be added to quote 14 (Renal replacement therapy in critically ill patients with acute kidney injury: 2020 nephrologist’s perspective. Nefrologia 2021 41:102-14 DOI: 10.1016/j.nefroe.2021.05.003). To include this point of view can act as a counter to the vincenza model to open minds in other regions and hospitals..

2.- I also suggest adding some more criticism to the fact that robust evidence is needed in relevant outcomes (such as survival after discharged) with new techniques (Oxyris i.e line 206).

3.- Finally, a brief reference to the pending research areas could be added to the conclusions.

Author Response

Interesting paper resuming the history and development of continuous therapies for AKI in CCU. Would be a reference paper for citation in future Reviews or position paper in this field. I consider as "special article" since it do no brings relevant results but interesting Review of the evolution of the technology tan Young generation tend to forget. Main author seems to be the most indicated person to do it due to his trajectory and experience in this field. I have really enjoyed discovering details that I had forgotten about this technique story.

Point 1.-Due to its structure as a narrative review, I have nothing to add in terms of methods and information. The reference to the need for integrated work between Nephrologists and CCU specialists seems especially relevant to me. This model is unfortunately not common to all countries, which is why other recent position papers of scientific society are especially relevant and could be added to quote 14 (Renal replacement therapy in critically ill patients with acute kidney injury: 2020 nephrologist’s perspective. Nefrologia 2021 41:102-14 DOI: 10.1016/j.nefroe.2021.05.003). To include this point of view can act as a counter to the Vicenza model to open minds in other regions and hospitals.

Reply 1. Thank you for the valuable comment. The manuscript has been revised accordingly (page 5, lines 139-143). We hope you find this suitable.

Point 2.- I also suggest adding some more criticism to the fact that robust evidence is needed in relevant outcomes (such as survival after discharged) with new techniques (Oxyris i.e line 206).

Reply 2. Thank you for the important comment. We added the limitation to the manuscript (page 6, lines 212-214). We hope you find this suitable.

Point 3.- Finally, a brief reference to the pending research areas could be added to the conclusions.

Reply 3. Thank you for the comment. We revised the conclusions accordingly (page 6, lines 228-233). We hope you find this suitable.

Reviewer 2 Report

The manuscript submitted in an excellent review on continuous renal replacement therapy. 

I suggest to include the reference on regional citrate anticoagulution for CAVH, Mehta et al, KI 1990.

https://pubmed.ncbi.nlm.nih.gov/2266683/

Author Response

Point 1. The manuscript submitted in an excellent review on continuous renal replacement therapy. I suggest to include the reference on regional citrate anticoagulation for CAVH, Mehta et al, KI 1990. https://pubmed.ncbi.nlm.nih.gov/2266683/

Reply 1. Thank you for the valuable comment. We added the reference (page 2, line 68).  We hope it is according to your idea.